# The Impact of Cycling Specialization on Successful Aging and the Mediating Role of Loneliness

**DOI:** 10.3390/ijerph19010019

**Published:** 2021-12-21

**Authors:** Haibo Tian, Wenting Zhou, Yajun Qiu, Yi Shang

**Affiliations:** 1School of Teacher Education, Shaoxing University, No. 508 Huan Cheng Xi Road, Yuecheng District, Shaoxing 312000, China; thbzhy@163.com; 2Department of Physical Education, Zijingang Campus, College of Education, Zhejiang University, No. 388 Yu Hang Tang Road, Hangzhou 310058, China; zipig12321@163.com (W.Z.); 12103034@zju.edu.cn (Y.S.)

**Keywords:** cycling specialization, loneliness, successful aging, mediating role, elderly adults

## Abstract

Recent studies have provided some evidence supporting that cycling specialization (CS) may be positively related to successful aging (SA) among elderly adults. However, there is a gap regarding the examination of the role of loneliness in the relationship between CS and SA. A hypothetical model was proposed to test the relationship between CS, SA, and loneliness. For this purpose, this study randomly conducted a questionnaire survey among 395 cycling participants over the age of 60 in China. The results showed that behavior, cognition, and affect had negative effects on loneliness. Behavior, cognition, and affect were positively associated with SA. Loneliness was negatively related to SA. Furthermore, behavior, cognition, and affect had positive and indirect effects on SA through loneliness. These results offered some new insights for understanding the relationship between CS and SA, especially considering the indirect effect of loneliness. The limitations and implications of the findings were discussed.

## 1. Introduction

SA has become a popular topic in gerontology research since Rowe and Kahn [1] proposed the distinction between “usual” and “successful” aging. It refers to three major components: low incidence of disease and disability, high level of cognitive and physical function, and positive involvement with life [1]. Over the past three decades, SA had been explained and modified from several different perspectives, resulting in an inconsistent definition of the concept [2,3,4]. At the same time, global population aging trends further aggravate and bring more inevitable challenges for SA. According to the results of the seventh Chinese National Census (CNC), the number of eldely adults who are 60 years old and over accounted for 18.7% (i.e., 264.02 million) of the total population by the end of 2020, which increased by 5.44% since the sixth CNC (2010). The increasing aging of the population has led the government to confront the difficulties of increasing social burdens, labor shortages, and the ethical problems of aging. Therefore, there is an urgent task for the government to adopt effective strategies to deal with this dilemma.

Loneliness was once viewed as an individual’s undesired negative experience, caused by a conflict between their wanted social relationships and those actually have [5]. It has developed into a significant and increasing public health issue since the arrival of an aging society. Existing literature supports that loneliness is a significant risk factor for bad health behavior, poor psychiatric condition, and physical health problems [6,7]. Previous studies have employed various interventions for loneliness, including training social skills, building a social environment, increasing social support, and a wider community approach [8,9]. However, the effectiveness of the interventions has not been robust, and further studies are required to deal with the challenges of how to offer and test such complex interventions [8,10].

Specialization can be best understood as an outcome or manifestation of a serious leisure pursuit [11]. Previous studies confirmed that specialization was highly and positively associated with serious leisure [12,13]. When elderly adults seriously engage in leisure sport activities, they will gain a series of durable personal and social benefits [14,15,16]. Recent research has verified that specialization positively contributed to individuals’ social relationships [17,18]. In addition, evidence also supports that specialization and loneliness can influence the quality of elderly adults’ aging life [6,19]. However, the existing literature has made little effort to explore the role of loneliness on the relationship between CS and SA. Considering the evidence highlighted above, this study aims to examine the relationship among CS, loneliness, and SA in China.

## 2. Literature Review

### 2.1. Cycling Specialization

The essence of “Specialization” is that outdoor recreation participants can be processed along a continuum from general interest and low involvement to specialized interest and high involvement [20]. It is usually used to explain the behavior and attitudes of leisure sport participants, and is reflected by equipment and skills using in the sport or activity setting [21]. During the past four decades, the concept of specialization has been widely used to explain people’s continuous progress as the time or intensity increases in various sport activities, such as fishing [22], mountaineering [23], diving [24], camping [25], cycling [26], climbing [27], and table tennis [28].

Previous studies have been devoted to how accurately measure specialization. The consensus was that the concept of specialization generally presented several multidimensional structures [19]. A three-system specialization loop proposed by McIntyre and Pigram [29] has received extensive support among these structures, which included behavioral, cognitive, and affective components. The behavioral system was characterized by frequency of participation and intensity of activities. The cognitive system referred to the knowledge and skills that an individual had accumulated about the activity. The affective system reflected the emotional factors related to enduring involvement (e.g., attraction, self-expression, and centrality). However, there still exists debate over which indicators make up the dimensions of the three-system specialization loop among different leisure activities [30,31]. In terms of cycling, recent research confirmed a better reliability and validity of the three-system structures of specialization under Chinese culture background [19].

Most scholars introduced the framework of specialization to segment participants into different groups, and compare within-group differences in their leisure preferences. For example, Song et al. examined the differences in hikers’ place attachment, demographics, past experiences, satisfaction, and intention to revisit according to degree of specialization (i.e., novice, affection-driven, and expert) [32]. In addition, other studies also explored the influence of specialization on other variables, including serious leisure [28,33], subjective well-being [19,34], leisure satisfaction [35] and so on.

### 2.2. Loneliness

Loneliness refers to a state of subjective feeling arising from an inconsistency between a person’s desired level or quality of social relationship and what they actually have [36]. Scholars had developed several different loneliness instruments from different perspectives. Most research has employed the one-dimensional UCLA (University of California, Los Angeles) Loneliness Scale, which had been confirmed with better reliability and validity among different cultural backgrounds [37,38,39]. In addition, loneliness has also been examined in a two-dimensional structure, comprising social loneliness and emotional loneliness [40]. Social loneliness was derived from the absence of a social network, while emotional loneliness resulted from losing a close emotional attachment. No matter which kind of structure for loneliness is used, it mainly reflects individuals’ subjective and negative evaluations in cognition [41].

The occurrence of loneliness is always accompanied by serious consequences for behavior, emotion, morbidity, and cognition. It is always related to increasing suicide risk [42], cognitive distortions [43], and elevated risk of Alzheimer’s disease [44]. A recent study claimed that loneliness may be caused by a chemical imbalance in the brain, but failed to explore the psychological mechanism of loneliness [45]. Furthermore, previous research also found that some factors are able to intervene with loneliness, including adaptability [46,47], a community development approach [48,49], and productive engagement [50].

### 2.3. Successful Aging

SA has not been unanimously defined, either from an objective perspective or in a subjective way [51]. The former opinion considered SA as a relatively low level of physical disability, high cognitive and physical functional capacity, and active engagement with life [1]. However, this definition ignored that aging was a kind of constructed reality that should be expressed by the elderly person’s subjective perception [52,53]. From a positive psychological orientation, Lee et al. defined SA as a structure with a relatively high quality of physical, psychological, social, and spiritual well-being adaptations [51]. They also developed a Chinese version of the Scale of Successful Aging, and confirmed its ideal applicability.

SA can contribute to understanding what people can do to promote their quality of experiencing good health as they get old [54]. Existing literature had made great efforts to define SA and explore the determinants of SA [2,55]. However, studies have paid little attention to exploring the influential factors of SA in the context of leisure activities. According to Silverstein and Parker [56], participation in meaningful leisure activities has been viewed as an important component for achieving successful old age. In a recent study [19], scholars confirmed that CS can positively influence elderly adults’ subjective well-being.

### 2.4. Relationship between CS, Loneliness, and SA

The social relationship formed in the process of participants’ leisure career may related to CS. When individuals are seriously involved in their leisure pursuits, they will gain more chances to improve the level of their social interaction and belonging through participating in the daily activities of various social groups [14,17]. Scholars also confirmed significant durable social benefits among different leisure sport activities such as long-distance running [57], cycling [19], and pickle ball [58]. In addition, based on an investigation of elderly baseball players, a recent study revealed that positive social influences were positively related to specialization [18]. Therefore, the dimensions of CS may influence the level of loneliness among elderly adult participants.

Continuous involvement in leisure activities can provide elderly adults with more opportunities to fulfil “lifelong friendship and belonging”, “pure happiness and passion”, “mental and physical health”, and “a sense of commitment” [14]. Evidence has suggested that specialization also makes a great contribution to increasing the elderly adults’ level of subjective well-being [19,59]. Furthermore, CS tended to fit well with the conceptualization of SA provided by Lee et al. [51], because it can offer a great opportunity to achieve physical or psychological well-being, improve social relationships, and express a good state of spirit. In other words, the benefits they gained during cycling are crucial to SA. Therefore, dimensions of CS may affect elderly adults’ SA.

A recent review showed that loneliness has been viewed as a significant important risk factor for poor health behavior, physical health conditions, and psychiatric problems [6]. Specialization was sometimes used to segment participants into different group (e.g., from novice to veteran) according to the variation in level of behavior, cognition, and affect [32,60]. As the degree of specialization progressed, individuals were more likely to become a member of a “social group” that possessed the same value, practice, and interest [61,62]. It can be beneficial for them to gain durable social benefits, such as social attraction, group accomplishment, and group development [63], which may contribute to decreasing the level of an individual’s loneliness and indirectly improve the elderly adults’ quality of life. In summary, loneliness may negatively influence SA, and has a mediating effect between dimension of specialization and SA.

### 2.5. Research Hypotheses

In this study, we aimed at examining the relationship between CS, loneliness, and SA among Chinese elderly adults. Based on the ideas mentioned above, we proposed a research hypotheses model that is shown in Figure 1. Hypothesis 1 to hypothesis 10 are as follows:

**Hypothesis** **1** **(H1).***Behavior is negatively associated with loneliness*.

**Hypothesis** **2** **(H2).***Cognition is negatively associated with loneliness*.

**Hypothesis** **3** **(H3).***Affect is negatively associated with loneliness*.

**Hypothesis** **4** **(H4).***Behavior is positively associated with SA*.

**Hypothesis** **5** **(H5).***Cognition is positively associated with SA*.

**Hypothesis** **6** **(H6).**
*Affect is positively associated with SA.*


**Hypothesis** **7** **(H7).***Loneliness is negatively associated with SA*.

**Hypothesis** **8** **(H8).***Loneliness has a mediating effect between behavior and SA*.

**Hypothesis** **9** **(H9).***Loneliness has a mediating effect between cognition and SA*.

**Hypothesis** **10** **(H10).**
*Loneliness has a mediating effect between affect and SA.*


## 3. Materials and Methods

### 3.1. Measurements

The Cycling Specialization Scale (CSS), modified from precious studies [29,32,64], was used to measure the degree of specialization for cycling participants. The CSS was proved to have good reliability and validity in a recent study [19]. There are three dimensions in the scale, namely “Behavior”, “Cognition”, and “Affect”. The CSS includes 9 items: behavior (3 items), cognition (2 items), and affect (4 items). The behavior dimension included three items reflecting recent cycling experience, lifetime cycling experience, and cycling strength. The cognition dimension consisted of two items addressing cycling knowledge and skill level. Both the behavioral and cognitive dimensions are graded on a 5-point Likert scale ranging from 1 (novice) to 5 (expert). The affect dimension comprised four items adapted from a previous study [29] and was rated on a 5-point Likert scale ranging from “disagree strongly” (score 1) to “agree strongly” (score 5). The statement for affect was as follows: “I find that a lot of my life is organized around cycling”. Cronbach’s alpha values ranged from 0.77 to 0.92, indicative of reliable consistency among the three subscales of CS.

The UCLA loneliness scale (Version 3) [39], which was revised on the basis of the previous studies [37,38], was used to measure the level of loneliness perceived by the elderly adults. The Chinese adaptation of the scale carried out by Gao et al. [65] was proven to have good reliability and validity. The scale includes 20 items, 9 of which are coded in reverse and 11 of which are straight. The statement for loneliness was as follows: ’How often do you feel that no one really knows you well’. The items are rated on a 4-point Likert scale ranging from “Never” (score 1) to “Frequently” (score 4). The loneliness scale showed better reliability in this study, and the Cronbach’s alpha coefficient was 0.87.

The Successful Aging Scale-Chinese version (SAS-C), which was developed by Lee et al. [51] under the background of Taiwan province, was used to measure the elderly adults’ subjective recognition of the quality of physical, psychological, social, and spiritual well-being. SAS-C included 20 items for four dimensions, including physical well-being (5 items), mental well-being (5 items), social well-being (5 items), and spiritual well-being (5 items). For example, the statement for physical well-being was “I can take care of my own daily life”. The items are rated on a 5-point Likert scale ranging from “extremely disagree” (score 1) to “extremely agree” (score 5). SAS-C was confirmed to have a good reliability coefficient in this study, ranging from 0.71 to 0.85.

### 3.2. Demographic Variables

Based on the experiences of previous researchers [19,43], a 6-item demographic variables form was developed by the researchers, which includes items such as sex, age, marital status, education, the number of siblings, and no major disease.

### 3.3. Procedures and Data Analysis

The data for this study were collected with the support of Shenyang Senior Cycling Association (SSCA). Questionnaires were randomly distributed during several different cycling activities from 15 August to 30 September 2021. Depp and Jeste have identified that 28 studies about SA recruited samples of seniors over 60 years old [66]. Therefore, a paper-and-pencil survey was performed among older cycling adults over the age of 60. Informed consent was signed by the elderly cycling participants before they answered the questionnaires. A total of 412 paper questionnaires were collected, but 17 of the respondents were excluded from the data analysis due to the age selection criteria (i.e., under 60 years old). Then, 395 questionnaires were used for examining the research hypotheses.

SPSS 23.0 was performed to analyze all data of this study. The demographic information, means, and standard deviations were calculated using descriptive analysis. Cronbach α was used to evaluate the internal consistency reliability of all the variables in this study. Pearson’s correlation technique was used to examine the correlation among behavior, cognition, affect, loneliness, and dimensions of SA. Multiple regression analysis was performed to examine hypothesis 1 to hypothesis 7. Process V3.3 developed by Hayes [67] was applied to check hypothesis 8 to hypothesis 10.

## 4. Results

### 4.1. Respondent Profiles

The respondents’ information are shown below, in Table 1. More than half of the participants were male (223 or 56.5%). Most of the participants were 66-and-above years old (231 or 58.5%). Most of the respondents were married (389 or 98.5%), and a smaller proportion of the participants were unmarried (2 or 0.5%) and divorced or widowed (4 or 1.0%). More than half of the respondents had a college or university education level (220 or 55.7%), and a smaller proportion one had a postgraduate education level (8 or 2.0%). A total of 87.3% of the participants had two or more siblings. In addition, 91.1% of the participants reported no major disease. Overall, the research sample was basically representative, except for unbalanced marital status.

### 4.2. Descriptive Statistics, Correlation Analysis, and Reliability

Table 2 shows the results of descriptive statistics and correlation analysis for each subscale. Affect had the highest mean score (*M* = 4.09, *SD* = 0.64) on the CS scale, followed by behavior (*M* = 4.08, *SD* = 0.82); cognition had the lowest mean score (*M* = 3.38, *SD* = 0.61). These results were basically consistent with findings of a recent study [19]. On the SAS-C scale, the highest mean score was found for the mental well-being (*M* = 4.32, *SD* = 0.22), followed by the social well-being (*M* = 4.22, *SD* = 0.26) and the physical well-being (*M* = 4.09, *SD* = 0.35); spiritual well-being had the lowest mean score (*M* = 3.92, *SD* = 0.52). These results suggested the participants usually evaluated their old-aged life positively. The respondents also reported low loneliness (*M* = 1.38, *SD* = 0.40), which indicated that cycling participants led a lively and warm life. In addition, the dimensions of SA increased as behavior, cognition, and affect increased (*r* = 0.15~0.77, *p* < 0.01); loneliness decreased as the dimensions of CS and SA increased (*r* = −0.24~−0.68, *p* < 0.01).

### 4.3. Direct Effect Testing

As shown in Table 3, three regression analysis models were performed to test research hypothesis 1 to hypothesis 7. Model 1 and Model 2 presented the standardized coefficients and significance using behavior, cognition, and affect as independent variables, loneliness and SA as dependent variables, and demographics information as control variables. The results indicated that sex (*β* = 0.087; *p* < 0.01) and age (*β* = 0.277; *p* < 0.001) were positively associated with loneliness, behavior (*β* = −0.057; *p* < 0.05), cognition (*β* = −0.162; *p* < 0.01), and affect (*β* = −0.465; *p* < 0.001) were negatively associated with loneliness; sex (*β* = −0.007; *p* < 0.05) and age (*β* = −0.309; *p* < 0.001) were negatively associated with SA, and marital status (*β* = 0.113; *p* < 0.01), behavior (*β* = 0.144; *p* < 0.01), cognition (*β* = 0.164; *p* < 0.01), and affect (*β* = 0.241; *p* < 0.001) were positively associated with SA. Thus, hypothesis 1 to hypothesis 6 were supported in this study.

Model 3 displayed the standardized coefficients and significance using behavior, cognition, affect, and loneliness as independent variables, SA as dependent variables, and demographics information as control variables. The results showed that marital status (*β* = 0.104; *p* < 0.01) was positively associated with SA, and age (*β* = −0.193; *p* < 0.001) and loneliness (*β* = −0.418; *p* < 0.001) were negatively associated with SA. Thus, hypothesis 7 was also supported.

### 4.4. Indirect Effect Testing

According to the results of regression analysis in Table 3, Process V3.3 developed by Hayes [67] was applied to examine the mediation effect of loneliness between independents (i.e., behavior, cognition, and affect) and dependent (i.e., SA). As shown in Table 4, loneliness had a significant effect as a mediator in the influence of the independent variables on SA. Specifically, behavior was partially mediated in the mediation analysis, while the full mediation of the others (i.e., cognition and affect) was determined. The standardized effect values were 0.109 (P1), 0.158 (P2), and 0.177 (P3).

## 5. Discussion

A proposed conceptual model was designed to examine the relationship between CS, loneliness, and SA among Chinese elderly adults. Results of multiple regression analyses and the Process V3.3 procedure supported all the ten hypotheses in this study. First, the findings showed that loneliness was significantly decreased as the dimensions of CS (i.e., behavior, cognition, and affect) increased. Study results extended the existing literature regarding the effect of specialization on loneliness in leisure sport activities. In general, loneliness was originated from the individuals’ imbalanced social relationship regarding what they expected and they really have [5,36]. When elderly adults progressed along the specialization continuum (i.e., from general to specialized), they will experience a lot of durable benefits of Stebbins’ serious leisure frame [12,33]. Then, those benefits can weaken the degree of loneliness in individuals’ aging life. A recent integrative review revealed that the most effective interventions were summarized as adaptability, productive involvement, and a community development approach [10]. Therefore, considering the lifestyle of the elderly, active cycling participation will be an important intervention for decreasing elderly adults’ loneliness.

Secondly, existing literature has shown that leisure sport participation revealed obvious characteristics of specialization in conjunction with SA [68,69]. This study fills the gap in the previous literature by examining the role of specialization on SA among cycling participants. The study results showed that SA was increased with the dimensions of CS (i.e., behavior, cognition, and affect). When elderly adults were seriously involved in leisure activities, they usually experienced higher levels of subjective well-being, which can contribute to a successful aging process [70,71,72]. Moreover, regular physical activities and exercise were important factors for elderly adults’ physical well-being [55]. They also gained lots of long-lasting outcomes, such as self-enrichment, social interaction, and self-gratification or pure fun [63], which can indirectly improve the level of their social well-being and mental well-being. So, the findings of this study suggest that strengthening the degree of CS can fully improve the level of SA in elderly people.

Third, extending to existing studies [55,66], this study confirmed that the degree of SA increased as loneliness decreased. Elderly adults were especially vulnerable to loneliness due to retirement, the death of spouses and partners, deteriorating health condition, and lack of confiding relationships. A recent review generalized that loneliness was an obvious risk indicator for lots of poor conditions such as health behavior, physical function, and psychiatric state [6]. Another study also verified that age differences and loneliness can accelerate the rate of physiological decline with age [73]. However, existing studies did not provide more robust strategies on the effectiveness of loneliness interventions [8,10]. Previous studies confirmed obvious social benefits from long-distance running, cycling, or pickle ball [57,58]. Hence, it is very important to improve the level of the elderly adults’ social relationships, especially through active leisure sport activities.

Finally, consistent with previous studies [56,72], this study confirms that serious engagement in leisure activities is an effective strategy for elderly adults to offset the social and physical deficits in later life. However, scholars have made less efforts to examine the role of loneliness on the relationship between CS and SA. This study enriches the existing literature by revealing that loneliness had an indirect effect on the relationship between behavior, cognition, affect, and SA. In other words, the dimensions of CS can effectively decrease the degree of elderly peoples’ loneliness, and indirectly improve the degree of the elderly peoples’ SA. Therefore, it is necessary for the government department or sport association to build a better leisure sport atmosphere, increase more social opportunities, and attract more elderly adults to become involved in various leisure sport activities.

## 6. Conclusions

This study aimed to explore the relationships between elderly adults’ CS, loneliness, and SA. Our studies findings filled the gaps in previous studies, and contributed to understand the link between CS and SA. Compared to the existing literature, this study extended previous findings about the role of CS on SA. It revealed that CS negatively affects loneliness, and loneliness negatively influenced SA. Furthermore, the study results also confirmed that loneliness had an indirect effect on the relationship between behavior, cognition, affect and SA. Considering the accelerating aging society, local government departments and sport associations should develop a scientific fitness plan and carry out a variety of leisure sports activities to further promote the quality of elderly adults’ life.

From a theoretical perspective, the findings support the positive influence that active leisure sport activities can have on experiencing a successful aging life. These results also extended the existing literature by testing the mediating effect of loneliness on the relationship between CS and SA. In addition, these findings are beneficial to understanding the predictors and influential mechanisms of SA. In terms of the practical standpoint, being seriously involved in leisure sport activities can offer elderly adults lots of durable health-related benefits, which can obviously help elderly adults to cope with the loneliness after retirement. Therefore, government departments should fully consider the physical and psychological characteristics of the elderly, spare no efforts to establish feasible programs, and widely promote leisure activities to cater the elderly adults’ need of pursuing a better life.

Despite the several implications mentioned above, this study should be interpreted cautiously. First, this study randomly recruited elderly adults from a sample of SSCA, and the sample was not fully representative of the cycling population in China. Future study should investigate a wider geographic area and be extended to various leisure sport activities. Second, SA was measured from a subjective perspective in this study, but other scholars have also tried to define SA either in an objective way or as a comprehensive opinion [1,2]. It is essential to examine related issues in the future with the use of more applicable measurements. Lastly, this study mainly examined the influence of loneliness in the relationship between CS and SA. However, other variables such as social support or satisfaction with life, may have an indirect effect on the role of CS on SA. Therefore, it is recommended for future study to explore the influential mechanism of SA.

## Figures and Tables

**Figure 1 ijerph-19-00019-f001:**
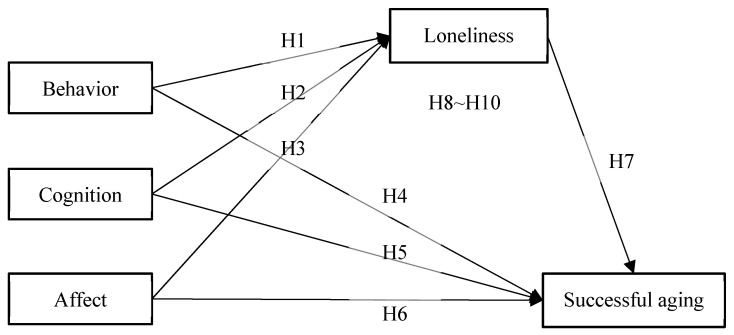
Proposed conceptual model.

**Table 1 ijerph-19-00019-t001:** Respondent profiles (*n* = 395).

Characteristics	Frequency (*n*)	Percentage (*%*)
*Sex*		
Male	223	56.5
Female	172	43.5
*Age*		
61–65	164	41.5
66–70	86	21.8
71–75	68	17.2
76 and above	77	19.5
*Marital status*		
Unmarried	2	0.5
Married	389	98.5
Divorced or widowed	4	1.0
*Education*		
High school or less	167	42.3
College or university	220	55.7
Postgraduate	8	2.0
*Number of siblings*		
1 and below	50	12.7
2–3	243	61.5
4 and above	102	25.8
*No major disease*	360	91.1

**Table 2 ijerph-19-00019-t002:** Measurement model results.

Variables	*M*	*SD*	*a*	1	2	3	4	5	6	7	8
1 BEH	4.08	0.82	0.77	1							
2 COG	3.38	0.61	0.92	0.70 **	1						
3 AFF	4.09	0.64	0.89	0.71 **	0.77 **	1					
4 PHW	4.09	0.35	0.71	0.64 **	0.65 **	0.72 **	1				
5 MEW	4.32	0.22	0.74	0.15 **	0.21 **	0.22 **	0.35 **	1			
6 SOW	4.22	0.26	0.85	0.37 **	0.45 **	0.53 **	0.52 **	0.25 **	1		
7 SPW	3.92	0.52	0.82	0.67 **	0.69 **	0.77 **	0.81 **	0.28 **	0.65 **	1	
8 LON	1.38	0.40	0.87	−0.57 **	−0.57 **	−0.60 **	−0.64 **	−0.24 **	−0.54 **	−0.68 **	1

Note: BEH = behavior; COG = cognition; AFF = affect; LON = loneliness; PHW = physical well-being; MEW = mental well-being; SOW = social well-being; SPW = spiritual well-being; *** p*
*<*
*0.01.*

**Table 3 ijerph-19-00019-t003:** Main regression results on loneliness and successful aging.

Variables	Model 1: LON		Model 2: SA		Model 3: SA	
	*β*	*p*	*β*	*p*	*β*	*p*
Sex	0.087	0.004	−0.007	0.039	−0.041	0.253
Age	0.277	0.000	−0.309	0.000	−0.193	0.000
Marital status	−0.020	0.497	0.113	0.002	0.104	0.003
Education	−0.039	0.190	0.034	0.348	0.018	0.597
Number of siblings	0.005	0.876	0.067	0.093	0.069	0.066
BEH	−0.057	0.012	0.144	0.008	0.119	0.025
COG	−0.162	0.001	0.164	0.007	0.096	0.096
AFF	−0.465	0.000	0.241	0.000	0.047	0.468
LON					−0.418	0.000
F	100.797		47.345		52.684	
*p*	<0.001		<0.001		<0.001	
R^2^	0.676		0.459		0.552	

Note: BEH = behavior; COG = cognition; AFF = affect; LON = loneliness; PHW = physical well-being; MEW = mental well-being; SOW = social well-being; SPW = spiritual well-being; SA = successful aging.

**Table 4 ijerph-19-00019-t004:** Indirect effect of independent variables on successful aging.

Analysis Paths	Sta. Effect	BootSE	BootLLCI	BootULCI	*p*
P1 BEH → LON → SA	0.109	0.017	0.079	0.145	Sign.
P2 COG → LON → SA	0.158	0.025	0.111	0.208	Sign.
P3 AFF → LON → SA	0.177	0.029	0.119	0.234	Sign.

Note: BEH = behavior; COG = cognition; AFF = affect; LON = loneliness; SA = successful aging.

## Data Availability

The data presented in this study are available on request from the corresponding author. The data are not publicly available due to restrictions i.e., privacy or ethical.

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
