# Peer review of "The Impact of Cycling Specialization on Successful Aging and the Mediating Role of Loneliness"

_ijerph, 2021, doi:10.3390/ijerph19010019_

Round 1

Reviewer 1 Report

Suggestion: Major revision

  1. Please re-write the assumptions as a paragraph clearly outlining what methods you used for testing that hypothesis. You may consider starting the paragraph with, “ In this study, we aimed at….” to frame your hypothesis and aims. The figure (figure 1) is good and gives the reader about your vision for the paper. But these are usually mentioned in paragraphs and not in numbered points. Kindly change them accordingly.
  2. Please include a detailed paragraph on age of subjects, gender ratio of subjects, mean/median ages and if they had any age related condition/disease. This is crucial in understanding successful aging. Then refer this to table 1. Did you consider any exclusion criteria for the sample, for example, was any of the subject experiencing any kind of health implications? Just a cut off age of <60 is insufficient data.
  3. Did you make the sign any consent forms? Need any ethics for this study at all? None of that seems to be mentioned in the paper
  4. Is there a possibility of knowing what kind of questions were asked in the questionnaire to assess the parameters mentioned? The methods/models used also need to be explained in further details for further clarity.
  5. The results section need critical analysis of the data. Currently it appears as statements bunched together. It needs to be more critical and authors need to challenge themselves to strengthen their data
  6. Discussion needs major revision and strengthening. Quite a few of the results confirmed what previous studies had already found. How do you suggest that your study is novel? Each of finding needs to be discussed critically in discussion.
  7. Grammar needs a major check as almost every sentence has grammatical errors. For example, multiple use of ‘existed literature’ instead of ‘existing literature’.

Author Response

    We thank the reviewers for their valuable feedback that greatly helped us to improve the quality of our manuscript. The reviews contained many useful comments and corrections, particularly related to three major concerns: (1) there needs to be more descriptive information on the study variables; and (2) the paper needs to provide greater clarification and precision about the interpretation of proposed model, methods, and discussion; and (3) there needs to correct some grammatical errors. Here are the major changes we have made: (a) adding a more precise description on measurement; and (b) adding details to improve the research model, methods, results, and, discussion, and (c) We corrected all the sentences’ grammatical errors through language editing services. In addition, we clarified several additional issues raised by the reviewers; Our revised manuscript is attached, and a summary of our responses to the comments follows below. We have highlighted the changes made in the manuscript for your easy perusal. Our study has greatly benefited from your detailed and constructive feedback and we look forward to hearing from you soon.

Reviewer 2 Report

It would be helpful if you would explain the 3-system-loop in more detail.

You use the three factors Behavior, Cognition and Affect in your hypotheses. Expecially these three should be clarified in the literature review.

METHODS:

In the description of the methods it would be good to have one sentence to each scale that describes in a short overview what this scale is measuring.

Were there special criterions for inclusion regarding the intensity of cycling (for example hours of training per week?)?

RESULTS:

The results could be more explained in detail in the discussion - like what does the result mean in other words (not correlation coefficients or statistical numbers) and in practice. 

Author Response

    We thank the reviewers for their valuable feedback that greatly helped us to improve the quality of our manuscript. The reviews contained many useful comments and corrections, particularly related to three major concerns: (1) there needs to be more descriptive information on the study variables; and (2) the paper needs to provide greater clarification and precision about the interpretation of discussion. Here are the major changes we have made: (a) adding a more precise description on measurement; and (b) adding details to improve the research discussion. Our revised manuscript is attached, and a summary of our responses to the comments follows below. We have highlighted the changes made in the manuscript for your easy perusal. Our study has greatly benefited from your detailed and constructive feedback and we look forward to hearing from you soon.

Round 2

Reviewer 2 Report

I replied in your text - please see the text in blue colour. 

Author Response

Thans for you kind suggestion. We made the adjustment on the submmit paper. You can see the changes on the cover letter.
